# From Protosolar Space to Space Exploration: The Role of Graphene in Space Technology and Economy

**DOI:** 10.3390/nano13040680

**Published:** 2023-02-09

**Authors:** Tanya Scalia, Lucia Bonventre, Maria Letizia Terranova

**Affiliations:** 1Italian Space Agency (ASI), Technology Unit, Via del Politecnico, 00133 Rome, Italy; 2Italian Space Agency (ASI), Legal Affairs Unit, Via del Politecnico, 00133 Rome, Italy; 3Department of Chemical Sciences and Technologies, Tor Vergata University of Rome, Via della Ricerca Scientifica, 00133 Rome, Italy

**Keywords:** graphene, nanocomposites, materials for space, space-related applications, space economy, patents and scientific literature trends, technology intelligence, solar sails, antennas

## Abstract

This paper aims to analyse the state-of-the-art of graphene-based materials and devices designed for use in space. The goal is to summarise emerging research studies, contextualise promising findings, and discuss underway strategies to address some specific space-related problems. To complete our overview of graphene-based technology and address the relevance of graphene in the wide scenario of the space economy, we also provide an analysis of worldwide patents and the scientific literature for aerospace applications in the period 2010–2021. We analysed global trends, country distributions, top assignees, and funding sponsors, evidencing a general increase for the period considered. These indicators, integrated with market information, provide a clear evaluation of the related technology trends and readiness levels.

## 1. Introduction

It is certainly not surprising that a material such as graphene has found, and is increasingly finding, wide application in space. Indeed, graphene is 200 times stronger and 100 times more tear-resistant than steel. Moreover, it is the world’s best conductor of electricity and heat, it is characterized by flexibility, transparency, stability, and impermeability, and it has the highest melting point of any material in vacuum. All these features optimally match the needs of current and forecasted space missions, since the latter require materials that are able to withstand extreme conditions of heat, radiation, and impact without losing their functional integrity [1].

The materials employed in the components and structures of a space mission experience strong stresses during the pre-launch ground activities of qualification and testing. During the lift-off and launch phases, the materials are subject to strong vibrations, shock waves, changes in gravitational forces, and high thermal gradients. When in orbit, the materials face adverse effects due to the space vacuum, cosmic radiations, solar wind, atomic oxygen, plasma charging, micrometeoroids, and space debris. Materials of a spacecraft in orbit around the Earth are exposed to temperatures in the −180 °C/+180 °C range, those close to other planets are exposed to temperatures as low as −200 °C, and those in the vicinity of the Sun are exposed to temperatures higher than 520 °C. The re-entry from space missions exposes the materials to overheating. Under the action of so many adverse processes, most conventional materials undergo a degradation of their structural and functional properties. Here, the new, leading, exciting artificial material graphene plays a relevant role: since its discovery, it has demonstrated huge capability in solving many of the problems arising in space-related applications.

What is generally unnoticed is that either astronomical observations of interstellar dust and planetary atmospheres, or the analysis of comet, asteroid, and meteorite matter, have revealed that this artificial high-tech material is a naturally-formed nanostructure [2], representing ~1.9% of total interstellar carbon [3,4].

Simulation experiments have provided evidence of the chemical steps and physical pathways leading to the formation of two-dimensional sp2-carbon nanostructures in deep space. The current hypothesis is that graphenic structural units originated from a protosolar carbon reservoir and were synthesized in a high-temperature zone near the proto-Sun and during the solar system’s earliest era [5,6]. Most likely, these hexagonal honeycomb structures were generated by the shock-induced decomposition of hydrogenated amorphous carbon grains, which should be very abundant in the circumstellar shells of dying stars. The evolution of such pristine carbonaceous masses in the form of solid units without a long-range order went on through a sequence of intermediate sp^2^ configurations, which resulted either in crystalline graphite or in the building blocks of amorphous sp^2^ carbons, the so-called black carbon [7,8].

The studies by Novoselov et al. [9] highlighted that the two-dimensional graphene couples contained the ultimate low mass density with a high stiffness (1 TPa), a great tensile strength (130 Gpa), and high electrical (350,000 cm^2^/V s) and thermal conductivity (5.3 kW/m K). Unfortunately, such outstanding properties are exhibited only by high-quality single-layer graphene, a miracle material that, up to now, can be produced only in small amounts by using sophisticated methods [10].

The unusual combination of mechanical, electrical, electronic, optical, and thermal properties of graphene have led to a focus on great efforts in the set-up of techniques for the mass-production of graphene-like nanostructures that are able to offer solutions to a variety of technological problems [11,12]. Presently, a lot of physical methods and chemical approaches allow the large-scale production of few-layers graphene, graphene platelets, graphene quantum dots, and reduced graphene oxide (rGO). This last material results from the chemical reduction of graphene oxide (GO), which is conducted by chemical exfoliation and the oxidation of crystalline graphite [13,14].

Even if characterised by lower performance with respect to conventional graphene, the achievements obtained using this graphene for the engineering of advanced materials, structures, and devices now make these nanomaterials a key player in all the space-related sectors.

Therefore, whereas graphene detected in interstellar media and ancient astromaterials is used to re-tell the story of the universe, artificial honeycomb carbon structures (graphene-based solids) go to space as cutting-edge materials that are able to enhance the mechanical, chemical, electrical, and optical properties of the components and structures in spacecrafts and satellites. In space domains, such multifunctional lightweight graphene materials (hereinafter referred to as graphene) have a transversal impact across all the sectors, from the processing of materials to the assembly of ground equipment and the launcher industry, and from satellite manufacturing/services to space science and exploration, just to name a few.

However, advanced and highly competitive coatings, structural components, or functional systems rarely make use of graphene on their own. For space applications, graphene is used mainly in combination with metals or polymers, giving rise to nanocomposites where the graphene insertion greatly influences the features of the host matrix. The engineering of graphene-based composites and the design of materials with the on-demand combination of tailored properties are therefore a primary goal of today’s space-related technologies.

The focus of this paper is to describe the state-of-art of graphene-based composites and to briefly illustrate some interesting examples of how graphene is applied to overcome challenges posed in space, dwelling on certain key themes that require additional research and drawing attention to some unexpected and surprising applications. Moreover, the issue of the space-related economy [15] is also addressed. In this context, we felt it worthwhile to highlight the relationship between technological evolution and the space economy generated by graphene. This last topic, described in Section 4, has been covered by analysing the worldwide patents and scientific literature trends in the period 2010–2021.

## 2. The Engineering of Graphene

### 2.1. Metal-Based Nanocomposites

The aircraft and spacecraft industries rely heavily on aluminium, magnesium, and titanium alloys due to their light weight, high mechanical properties, low cost, and also reasonably good temperature resistance [16]. Graphene insertion inside the metals greatly improves several fundamental mechanical properties, such as the ultimate tensile, compressive, and rupture strengths, elongation at failure, Young’s and shear moduli, and density. In this view, the graphene-based nanocomposites meet precisely the demands of the Federal Aviation Administration (FAA) reported in the Metallic Materials Properties Development and Standardization (MMPDS) report [17]. Beyond its crucial role in the fabrication phase, graphene was demonstrated to be effective also as an additive in thermal spray metallic coatings for the fast repair of aerospace engine components. In particular, the mixing of multi-layered graphene with Ni-Al powders was found to decrease the structural defects in repair coatings, improving hardness and tensile adhesion strength as well as reducing residual strain and the coefficient of friction [18].

The wide range of different uses envisaged for different environments needs the fine tuning of highly specific, and sometimes rather complex, materials. An example is given by graphene layers, functionalised with carboxylic and hydroxyl groups, wrapping the silicon carbide (SiC) particles used as the filler in microstructured aluminium. These nano composites showed an increase in storage modulus and ultimate tensile strength by a factor of ~2.1 and ~3.4, respectively, compared to pure aluminium [19].

The improvement of nanocomposite mechanical features is accompanied by the improvement of thermal properties, such as thermal conductivity, specific heat capacity, and the thermal expansion coefficient. These effects can be of great value, as can be appreciated from the results obtained in the case of graphene–magnesium nanocomposites, a material much lighter than any existing alloy used by the aerospace industry [20].

However, the achievement of outstanding thermo-mechanical results is not an easy task to obtain just by adding graphene to a metal because the insertion of a nanofiller produces structural modifications in the host metal lattice. To obtain a material with the required properties, a strict control for the material structure is needed at the nanoscale level. An example of how to manage this issue can be found in [21]. In this paper, the interface between graphene and copper (Cu) has been deeply analysed by Molecular Dynamics simulations in order to optimise the mutual interactions of the two components. Depending on the texturing of Cu along the (1 0 0), (1 1 0), and (1 1 1) crystallographic orientations, the calculations predicted different melting temperatures for the graphene/Cu hybrids. Also, dramatically different values of mechanical strength were evaluated from the stress-strain curves obtained by applying uniaxial tensile loading along the different directions dictated by the graphene edges. Anjam et al. [21] found that, along the armchair direction, the mechanical strength of graphene/Cu (1 0 0), (1 1 0), and (1 1 1), compared with pure Cu, increased by about 380%, 370%, and 450%, respectively, whereas along the zigzag direction, the strength increased by about 1200%, 1000%, and 1480%. These differences were related to the diverse lattice mismatches generated by graphene contacting different Cu planes and to the occurrence of glide and shuffle dislocations. The graphene edges were found to drive the propagation of such dislocations, and to indirectly influence the thermal stability and the mechanical strength of the composites.

Experimental nanostructural studies on graphene-Al nanocomposites for aerospace applications have confirmed the Molecular Dynamics (MD) simulations [22]. The enhancement of the mechanical properties exhibited by the graphene-reinforced metal have been ascribed to the lamellar fine grain structure of the nanofiller. Such features minimise point defects, surface defects, and line dislocations of the host structure. Images of single-atomic-thickness graphene embedded in three different grades of aluminium alloy are shown in Figure 1.

The strengthening of the composite material is achieved by the transfer of the load from the matrix to the graphene flakes, which, in this view, would act also as a load-bearing component and not just like a control for the dislocation movements. The effective transfer of the strain to the filler enables the processing of Al/graphene powders via the traditional metallurgical routes. The structural components of the external fuel tanks of launch vehicles are fabricated using a microwave powder-sintering method followed by hot extrusion [22].

Overall, the most recent research has underlined how the thermo-mechanical properties of graphene-metal materials are driven by their structural features at the nanoscale level, confirming how nanocomposites for aerospace applications must be accurately designed.

### 2.2. Polymer-Based Nanocomposites

Graphene-based polymer nanocomposites are currently at the forefront of materials used for aerospace structural components. The obvious mechanical advantages that result from the ingenious coupling of graphene with a suitable polymer have indeed made such light-weight and easily manufactured nanocomposites a standard in the engineering of aerospace structural components. However, the many outstanding properties that graphene transfers to the host polymer have also been exploited in other applications, often still at the research stage, such as those in thermal management or space propulsion [23].

The large number of polymers under consideration and the many actual or planned space applications need to put into play all the techniques/protocols settled until now to produce, engineer, and process multifunctional nanocomposites [24]. One of the main tasks is the production of nanostructured flexible films with given optical and electromagnetic properties. This task needs a strong control, at the nanoscale level, for the geometry and the aggregation state of graphene nanoplatelets, and moreover for their dispersion inside the polymeric matrix [25]. For some applications, such as in gas-permeation barriers, the number of graphene layers coating polyurethane films has also been found to play a fundamental role [26].

In this context, the challenge is to develop manufacturing techniques and to envisage other hitherto unexplored approaches that are able to assure the specific performance required in uses that include, among others, sensors, antistatic coatings, and electromagnetic interference shields [23].

A further space-specific application of graphene-based nanocomposites is the fabrication of ablative coatings in the hyper-thermal environments experienced by space vehicles and rocket motors. Ablation tests demonstrated that the addition of graphene to elastomeric matrices greatly reduces the temperature rise at the back face of the coatings, highlighting an increase in thermal stability and in the heat absorbance capability of the nanocomposites with the increasing of the filler-matrix ratio [27]. The main results of this study are shown in Figure 2.

Approaches based on multi-scale methods are also widely employed in the case of graphene/polymer nanocomposites to model the structure of the materials and the architecture of the systems following the requirements of the specific spatial uses [28]. To design new graphene-based nanocomposite layers for aerospace applications, the synergistic coupling of multi-scale modelling with experiments has proven to be a successful approach [29]. This paper describes the Integrated Computational Materials Engineering (ICME) method that has been applied in fabricating large aerospace laminate structures made by graphene/carbon fibre/polymer. This new composite has been designed in the frame of the Composite Exploration Upper Stage (CEUS) NASA project and is used for the forward skirt structure of the Space Launch System. The nanocomposite material, used as the face sheet of the sandwich panels in the barrel section of launch vehicles, improved the resistance to open-hole compression failure in the structure. Moreover, due to the graphene insertion, the panels showed a 22% reduction in weight with respect to the conventional composite ones.

To deeply understand the graphene-induced effects on the mechanical properties of polymers used for aerospace structures, in particular for sandwich panels, a major topic is the precise measurement of graphene-polymer interfacial strength. An interesting approach used to quantify such interactions is illustrated in [30]. Here, the interfacial strength of graphene and of oxidized graphene with a poly-epoxy resin matrix are measured using strain sensors pasted on the assembled panels.

It is noteworthy to report that a vehicle made with graphene-polymer composites has been assembled by Orbex, the UK-based private, low cost orbital launch services company [31]. The two-stage rocket Orbex Prime, unveiled in May 2022, has been designed to carry up to 150 kg of payload into a Sun-synchronous orbit. The main structures and the fuel tanks of Orbex Prime are built using polymer/carbon fibre/graphene composites. The design of the Orbex Prime body is shown in Figure 3.

## 3. Emerging Applications and Open Challenges

### 3.1. Thermal Management Systems

From the beginning of the “graphene era”, thermal management systems used in space have greatly relied on the use of the material. Even if, up to now, the value of thermal conductivity foreseen by Novoselov et al. [9] has not been experimentally demonstrated, the produced single-layer graphene reaches, in any case, a high thermal conductivityof 2.5 kW/m K [32].

In the recent years, the European Space Agency (ESA) have started to test graphene for the cooling of loop heat pipes in a low-gravity environment [33]. The main component of a loop heat pipe employed in aerospace and in satellite instruments is a metallic wick that transfers heat from the hot components to a coolant fluid. The coating of porous metallic wicks by graphene-based materials was also found to improve the efficiency of the heat pipe in low-gravity conditions, and the Graphene Flagship Collaboration is planning to utilise graphene for the cooling of ESA satellites.

The miniaturisation of the thermal management systems that can be achieved using graphene materials is of utmost importance for small satellites, which are contributing more and more to the advancement of scientific explorations and to the reduction of mission costs. The altitude reached by a small satellite in a low Earth orbit is between 15 × 10^3^ m (the so-called thermosphere) and 10^6^ m (the outer limits of the atmosphere below the van Allen radiation belts). Here, the satellites are exposed to solar radiations, Earth albedo, and Earth infrared radiations, effects that altogether contribute to a remarkable temperature increase. Temperature variations are currently kept under control in large satellites by loop heat pipes (active systems) and also by the passive protection offered by insulating paints or multilayer insulation blankets.

However, for small satellites, efficient thermal dissipation technologies are still under development. A new concept for heat dissipation in these systems is to adapt the loop heat pipes proposed for larger satellites, adding graphene nanoparticles to the conventional heat transfer fluids. Increased thermal performance induced in fluids by the graphene phase resulted in a continuous heat dissipation of 75 W, enabling the retention of the 6 U CubeSat payload within a suitable temperature range [34]. Furthermore, for small satellites, coupling the use of graphene as an additive in fluid loops with the use of aerogel in the insulator sheets has also been proposed [35].

### 3.2. Anti-Wearing and Anti-Corrosion Systems

For all spacecrafts, rockets, and satellite components, corrosion is a very big challenge to deal with. Prolonged exposure to a variety of adverse agents in the whole of space, from near-Earth to deep space, produces material loss in the surfaces. This effect is especially evident in aluminium alloys, and the coating by anticorrosion layers is therefore a pivotal subject of applied research. In this context, GO-based materials inserted in a polymeric matrix were demonstrated to improve the corrosion resistance of the applied layers (Figure 4), likely due to the more compact coating-substrate interface [36].

The exposure to erosive solar wind is an issue that hampers the use of conventional materials in the equipment employed in outer space. This problem arose particularly when the construction of a probe microscope-space satellite was planned. For such a high-precision device characterised by a reduced weight (not exceeding a few kilograms) and by the ability to work in vacuum, the need to counteract erosion was handled by using graphene multi-layered structures prepared by the thermomechanical compression of several sheets of CVD-produced single-layer graphene. This material has been plasma-treated in-laboratory under conditions imitating solar wind. The experiments confirmed that such light multi-layered structures are effective shields for equipment protection and also provide promising structural material for solar sailing [37].

### 3.3. Electronic and Photonic Devices

In the field of advanced optoelectronic systems and devices, graphene is now a rather ubiquitous material. Several reviews, to which the reader can refer in [38,39,40], illustrate the numerous applications, either commercially available or still visionary, proposed for this material. A recent paper by Sengupta and Hussain is especially interesting, since it describes in depth the “in progress” fields of batteries, touch screens, transparent memories, and integrated circuits [41]. In the following, we focus on some niche solutions designed for specific space applications.

Graphene sensors highly sensitive to THz frequencies that are able to detect polarisation and moreover to be adjusted electronically have been planned to be used in Sun THZ. This is a telescope designed to detect flares and phenomena of solar explosions that occur on the Sun’s surface, and cause high levels of radiation in outer space. Sun THz is an enhanced version of Solar-T, a double photometric telescope launched in 2016 by NASA in Antarctica in a stratospheric balloon. The Solar-T telescope captured, for the first time, the energy emitted by solar flares at two frequencies in the 3–7 THz range, corresponding to a segment of far infrared radiation. The new Sun THz telescope will be sent to the International Space Station (ISS), where it will remain, to measure the full spectrum of solar flares, which is from 0.2 THz to 15 THz. The unconventional detection of such frequencies, which can be measured only from space, together with the foreseen prolonged exposition of the telescope to high radiation levels, make it necessary to use materials with outstanding photonic properties and high radiation resistance. The graphene-based sensors for the Sun THz telescope are being presently developed in the frame of an international Brazil-Russia-UK collaboration [42].

Other types of sensors that take advantage of the use of graphene are bolometers, devices widely used in sub-millimetre astronomy to analyse radiations through the measurement of radiated heat. The small heat capacity, the weak electron-phonon coupling, and the low electrical resistance of graphene are intrinsic properties that can strongly improve the performance of such detectors. Du et al. [43] analysed the most recent theoretical and experimental achievements in the field of cryogenic graphene-based bolometers. Their study considered the phonon cooling mechanism and its dependence on temperature, the doping and disorder of graphene structures, and the various approaches used to realise bolometric detectors.

However, it must be noted that photonic devices, such as bolometers or saturable absorbers in ultra-high bandwidth detectors, need the manufacturing of large area graphene, either as a single layer or multi-layered. To that end, scientists at NASA developed, in 2019, a synthesis process that was able to produce graphene sheets that were suitable for the assembling of such photonic devices [44].

In addition, sensors based on the Hall effect are widely used in space. The Hall effect sensors are key electronic components present in a variety of devices that are used for proximity sensing, positioning, speed detection, and current measurements. The devices assembled with conventional semiconductors must be encapsulated in heavy-radiation-resistant packages to avoid damages and failures induced by neutron radiation. Tests performed in the UK’s National Physical Laboratory proved that graphene Hall sensors can withstand exposures to neutron doses as high as 241 mSv/h, about 30,000 times the neutron dose typically experienced in the International Space Station. Moreover, the replacement of silicon with graphene increases the sensitivity of these electronic devices, which require only few pW of power and are extremely lightweight. These valuable features make graphene Hall sensors well-suited for employment in satellites and spacecrafts [45].

### 3.4. Solar Sails

The solar-photon sails, commonly called solar sails, are innovative propulsion systems proposed to drive flights and the orbital manoeuvres of spacecrafts in deep space. A solar-photon sail utilises the pressure of solar photons to drive an object in space without using fuel or gases. The propulsion is given by the force that originates from light reflecting off a surface or being absorbed by it, and from the consequent pushing of the surface itself away from the light source. The concept of a solar sail is a potential option for driving long-duration extrasolar and interstellar exploration/colonisation missions, but the application requires a thin, light weighted, reflective, temperature tolerant, and space-environment-resistant sail. The technology of large sails made of polyimide and mylar coated with a metallic reflective thin layer was firstly tested in the IKAROS mission launched in 2010 by the Japanese Aerospace Exploration Agency (JAXA) and, after that, redesigned considering the use of graphene [46,47,48].

In [48], a low-weight solar sail obtained by the use of double-layer graphene on a holey copper grid is described. In this architecture, shown in Figure 5, the graphene sustains the highly-reflective ultrathin film covering the whole sail surface.

The sail system of graphene micromembranes supported by copper grids was tested under microgravity and vacuum conditions by measuring the displacement induced by laser irradiations of a 3 mm scale model. Lasers with optical DC powers of 100, 500, and 1000 mW produced forces in the 8–248 nN range, and demonstrated the feasibility of the light-induced acceleration of the 2D sails.

Currently, the visionary technology of solar sails made by free-floating graphene membranes has also been explored by China [49], by the Innovative Advanced Concepts Program of NASA [50], and by the Graphene Flagship Collaboration of ESA [51]. As reported in the ESA website [51], in 2020, a one-atom thin graphene solar sail under vacuum and in microgravity was successfully tested. The shining of a 1-W laser produced a 1-m/s^2^ acceleration of the sail. Solar sails based on graphene are extremely promising forms of spacecraft propulsion because these devices have few moving parts, use no propellent, and have long operational lifetimes.

However, a really innovative, cutting-edge graphene-based propulsion can be provided by a mechanism that is different from that of the photon-reflection, exploiting a new concept first put forward in a 2015 paper [52]. The authors demonstrated that properly assembled graphene sheets can produce the efficient light-induced emission of electrons ejected following an Auger-like path, and that the net momentum generated by this process can propel the bulk graphene material according to Newton’s laws of motion. The major breakthrough of this research was that the newly designed bulk graphene material maintained efficient light absorption, the easily achievable reverse saturation state, and the unique hot-electron relaxing mechanisms typical of single-layer graphene. This exciting finding pushed the research of China Academy of Launch Vehicle Technology towards the fabrication of graphene structures emitting energetic electrons under light illumination and therefore those that were suitable for use in spacecraft propulsion.

### 3.5. Antennas

Wireless communication systems in space need multiband operations and, at the same time, an optimal method for the utilisation of onboard components. In this context, many studies have focused on the fabrication of reconfigurable multiband antennas. However, there are other constraints on the actual applications of antennas in space, such as their low power, small size, and minimal cost. It has been demonstrated that graphene-based conductive inks offer the feasible realisation of antennas characterised by frequency reconfigurability and also by a significant reduction in cost, size, and maintenance. Indeed, the printing of this composite material enables the fabrication of textile-based, highly effective microstrip antennas capable of switching between the S-band (3.03 GHz, TM10 mode) and C-band (5.17 GHz, TM02 mode and 6.13 GHz, TM20 mode) [53].

More recently, Cherevko et al. [54] fabricated a flexible reflect array antenna in the GHz range, assembling various unit cells based on graphene. The printed graphene-based elements ensured mechanical stability, low weight, and reduced size, whereas the low-cost flexible reflect array antenna showed performance comparable with those of similar antennas, based on the use of metals [54,55].

Among graphene’s outstanding properties, the effective propagation of surface plasmon polariton allows graphene-based antennas to resonate at higher frequencies with respect to a metal patch antenna of the same size. The dual graphene patch antenna proposed in [56] enables to use a single antenna for two-way communication links in the Ka band. This result is very important for satellite applications, in which the disadvantages (narrow bandwidth, low gain, and losses) of metal patch antennas make their use rather difficult at high frequencies. Moreover, the miniaturisation of conventional antennas causes great attenuation, making challenging the manufacture and the implementation of wireless communication systems with reduced dimensions. An improvement in the performance of this type of antenna has been subsequently achieved by varying the chemical potential of graphene sheets. The increase in chemical potential caused a prominent increase in the values of the gain, and as such, graphene patch antennas were demonstrated to be highly effective for spatial operations [57].

Moreover, in small satellites like SmallSat and CubeSat, there is very little space to insert antennas as well as the solar panels providing energy, and the integration of the two devices in a single compact unit is a challenging task. An innovative approach to overcome this difficulty was proposed by Ram et al., who fabricated a micro-strip patch antenna integrated with a solar panel by screen printing a graphene-based nanomaterial [58]. This technique reduced, by one-third, the space occupied in the satellites by the traditional antennas, whereas the presence of graphene enhanced the optical efficiency of the solar panel and improved the antenna’s functionality in the 2.15–2.18 GHz band [58].

Graphene has several advantages over metals also at the THz band. It has a room temperature electron mobility of up to 230,000 cm^2^/Vs and an electrical resistivity of about 10^−6^ Ω·cm in the THz band. Such properties result in a loss lower than that of conventional metals. In addition, in graphene, the surface plasmons exhibit lower resonant frequencies than those in metals and extremely small wavelengths, and are tightly confined on the graphene sheet. The excellent physical properties of graphene at THz frequencies have been exploited in a series of graphene-based THz antennas as the reflect array formed by a circularly polarised source and several graphene-based unit-cells were proposed for operation in the THz regime. This design enables the realisation of a high-gain antenna with a large bandwidth and a phase tunability of a wide band (1.4–1.7 THz), making the system suitable for THz communications [59].

### 3.6. Ultra-Thin Lens

Another exciting use of graphene for space-related applications is in the fabrication of ultrathin flat lenses, a revolutionary technology that, in space, is hampered by the harsh environmental conditions. In this context, it was thought that graphene-based lenses could have better stability and could also maintain outstanding focusing performance under space conditions [60]. Using a direct femtosecond laser for the ablation of fully reduced GO films, a flat lens with a thickness of 150 nm and a focal length of 30 μm were fabricated. Figure 6 illustrates some of the applications foreseen of such a lens in space operations.

The stability of the graphene lens was tested in experiments simulating low-Earth-orbit space conditions, which include the exposure to extreme heat/cold cycles, strong UV, higher energy radiations, an ultrahigh vacuum, and atomic oxygen [61]. These ultra-light, ultra-thin flat lenses maintained their structural stability and their capability to achieve aberration-free high-performance images in a broad spectral range.

### 3.7. Energy and Powering

Electrical energy storage devices exploiting the good electrochemical features of graphene have been proposed since 2008 [62], and several graphene-based hybrid materials have been designed for supercapacitor components [63]. Presently, the supercapacitors are considered a new solution for space power supplies because they can overcome the main disadvantages of the lithium-based batteries [64]. These last devices, widely used as the main energy storage systems of satellites flying in low Earth orbits, face problems related to their mass and volume and also troubles when working under low temperature conditions. In this respect, the use of supercapacitors to complement conventional batteries and to assemble high power hybrid energy storage systems that are able to work in a wide temperature range has been proposed [65]. The new type of graphene-based devices, the so-called graphene supercapacitors (GSCs), demonstrated the feasibility to be used also as independent energy storage systems on-board spacecrafts.

An interesting example of GSC being specifically designed to be used without a backup of conventional batteries is reported in [66]. Here, GSCs were used as a power supply for the new type of electromagnetic separator systems in Q-SAT missions. The temperature and vacuum adaptability of the GSCs, along with their instantaneous high-power output, were found to match the best requirements of the electromagnets of the separation system, and these innovative supercapacitors guaranteed the unlocking and releasing of Q-SAT in flight on 6 August 2020.

## 4. Patent and Scientific Literature Analysis

In the previous paragraphs, we have examined many different applications of graphene-based materials in space. Most of the information provided above was extrapolated from scientific publications, contracts, etc. However, in this paragraph, we would like to lend a different perspective by providing some quantitative indicators of the scientific literature and patents. In fact, we are aware of how the statistical analysis of patent information and its visualization make a powerful and successful methodology for any competitive intelligence activity centred on technology [67], since the latter can be effectively used to monitor and evaluate technology activities. However, there are some known limitations of patent analysis, for example, the delay between the application and its actual publication, and also the fact that not all innovative activity is patented or even patentable (e.g., high costs, strategic decisions, the need to keep industrial secrets, etc.). This is why, in this paragraph, we integrate the patents’ indicators with those of scientific publications.

Thus, considering the information provided above, we hereinafter analyse the overall worldwide patents and literature to evaluate the technology’s evolution in the period 2010–2021.

By using two different dedicated databases, namely Orbit Intelligence [68] and Elsevier’s Scopus [69], we created our queries by combining graphene-related keywords with those of space technologies. The technologies associated with the use of graphene in space applications are expanding in a deep, pervasive, and rapid way, affecting the worldwide economy. Here, we explored the most recent evolutions (2010–2021) of both the patent and scientific literature data related to different graphene applications, as reviewed in the previous paragraphs. Patent indicators, complemented with the relevant scientific literature, may provide a clear overview of technology trends and allow the foreseeing of future developments. Although the number of patents is not in itself a sufficiently accurate indicator of the capability of producing new knowledge, data on patent publications, as well as scientific articles, are suitable for characterising the scientific and technological expertise of industries, research institutions, and academia, and to identify possible knowledge spill-overs. These types of indicators are also useful for the exploration of technology transfer and open-innovation opportunities.

On the one hand, patent-based indicators are extremely useful for comparing and monitoring trends in the technology output of different countries; here, this data is complemented with the scientific literature indicators to study empirical and theoretical works and capture what is known on the topics above. On the other hand, scientific literature represents the main source for communicating and disseminating the results of scientific research and, as such, it represents the record of the collective achievements of the scientific community over time.

In the patent analysis hereinafter presented, the data was analysed using the Orbit Intelligence Patent Database tool by Questel with the aim of extracting information from the patents’ titles, abstracts, claims, descriptions, and concepts to identify, among others:investment and scientific literature trends;country distributions of patents and literature;International Classification Classes (IPC4);top applicants by legal state;top affiliations;dependency of applicants by citation.

Insofar as the scientific literature is concerned, the following paragraph shows results obtained through Elsevier’s Scopus tool with the aim of extracting information from the titles, abstracts, and descriptions of scientific articles.

### Patents and Scientific Publication Resulting Trends

This paragraph presents the results obtained from the analysis of patent families (i.e., a set of patents taken in various countries to protect a single invention) and from the scientific literature published worldwide between 1 January 2010 and 31 December 2021. We focused our analysis on the keyword ‘graphene’ combined with keywords related to human exploration (e.g., ’astronaut’, ‘cosmonaut’) and long-term permanence (e.g., ‘deep space’, ‘spacecraft’). We excluded from our analysis keywords related to the launcher and to propulsion systems since they could be the object of a dedicated review. Our final dataset included 4967 scientific articles and 6531 patent families. In the following paragraphs, we examine the results of this dataset.

The graph in Figure 7 illustrates the evolution of applications over time in terms of scientific publications and protected patents, indicating the dynamics of the inventiveness of the portfolio studied. A significant increase in the number of both patent families and scientific literature can be observed. Such a trend can be explained by a significant interest in financing R&D activities with graphene in space. It is worthwhile mentioning that in the last two publication years, there has been a gap in patent information due to the 18-month delay between the filing of an application and its actual publication.

An exponentially-growing patent portfolio indicates that the applicants are in the phase of rapidly constructing their portfolios. A similar increasing trend can also be observed in the scientific publications, confirming the increasing efforts of the scientific community in applying graphene in space. The findings of the above review are fully confirmed by such results, which clearly demonstrate the wide interest in using graphene for space applications.

Following such a trend, the stabilisation of patent filings can be expected, which could be motivated by a stabilisation of R&D budgets–which leads to a flow of patent applications that is more or less constant without too much selectivity in applying for patents–and/or the desire to stabilise patent costs, which leads to significant selectivity in the filings and their maintenance.

The pie chart in Figure 8 illustrates the geographical distribution of both alive and dead patents protected at a national and European level. The leading countries are China (71%) and Korea (12%), followed by the United States of America (7%). However, other countries such as Japan, Taiwan, Germany, and Canada also demonstrate an involvement on this topic.

This graph also provides information on the commercial strategies of the applicants in the sectors studied as the national filings are a good indicator of the markets that need to be protected. In addition, many applicants protect the geographical areas where the manufacturing sites of their competitors are located.

The following picture (Figure 9) shows the distribution of the scientific literature by country, confirming that China is the leading country in terms of scientific publications worldwide for the topics under examination; however, many other countries like United States of America, Russia, India, and some European countries (e.g., Italy, Germany, and France) play a significant role.

The following graph (Figure 10) is based on International Patent Classification and shows the most frequent IPC codes (IPC4). These codes were established by the Strasbourg Agreement of 1971 and provide a hierarchical system of language-independent symbols for the classification of patents according to the different areas of technology to which they pertain. This graph is useful in identifying patents in a domain and in a field that may have multiple uses. In our findings, the most recurrent IPC code is H01M, “Processes or means, e.g., Batteries, for the direct conversion of chemical energy into electrical energy”.

The description of the IPC4 reported in the picture above is provided in Table 1:

The graph in Figure 11 illustrates the key affiliations in the group of patents analysed (~6500) according to their legal status, which is a good indicator of the level of inventiveness of the active players. This information makes it possible to identify applicants who have withdrawn their patents (abandonment, lapse, and/or expiration) and those who are still maintaining them (applications and patents granted still in force). Indeed, this data is useful for understanding the legal state of patents, and namely the percentage of “alive” (76% of the total amount) and “dead” (24% of the total amount) patents. For the sake of clarity, patents with no expiration events within their predicted term are considered “alive”. Patents that pass their expiration date (20 years) and those with a terminal event (e.g., failure to pay maintenance fees) are considered “dead”. A family remains alive as long as it contains at least one record with an “alive” status.

Figure 12 shows the main affiliations with the highest level of scientific publications for the period 2010–2021. As anticipated, the Chinese institutions (e.g., Chinese Academy of Sciences, Ministry of Education, and Tsinghua University) are highly active, however, other prominent research centres from different countries—e.g., Centre National de la Recherche Scientifique (France), the National Institute for Material Science (Japan), and Massachusetts Institute of Technology (USA)—also actively work on graphene related topics.

The following graph (Figure 13) illustrates citations between applicants. This information identifies portfolios that have strong interactions with each other. A portfolio that is strongly cited by most players is likely to be a pioneering or a blocking portfolio. This representation is also a good indicator of the applicants’ inclination to collaborate, and it identifies their preferred partners. The broadest citation dependency involves some of the main key applicants, such as Tsinghua University, Samsung Electronics, and Dalian Institute.

## 5. Concluding Remarks

The outstanding intrinsic properties of graphene make it an extremely suitable candidate as the material of choice to deal with the peculiar characteristics of out-of-Earth environments.

The use of graphene and graphene-like nanostructures in space is not limited to the applications briefly reviewed in this paper, but also include radiation shielding, fuel propellants, telecommunications, biomedical technologies, life support systems, and many more. As a whole, a variety of graphene-based products have already found their way into space but, despite great efforts, a number of challenges and untapped opportunities still remain unsolved for such a relevant nanomaterial.

The few examples of the forefront graphene-based technologies illustrated in this paper outline that there are still some goals that have not been realised, or at least not completely. The first issue concerns “true” high-performance graphene. To succeed in making graphene operative at its best requires the development of methodologies for the mass production of large-area crystals of one-atom-thick sp2-bonded Carbon. The ultimate performance of the future’s 2D “miracle material” would provide a solution for almost every present and future space-related technology. Several research groups worldwide are already working on this issue and the hope is that they could be only a few “paradigm-shifting” breakthroughs away from the goal.

As regards the less-performant graphene-like nanomaterials, leading voices in material science evidence a growing commitment to provide the mass production of few-layers graphene, graphene platelets, graphene quantum dots, reduced graphene oxides, and other graphene-related structures. Near-future research efforts will be aimed towards the large-scale production of structures with homogeneous size, shape, and thickness.

In this context, even if the potential toxicity of these amorphous materials has not been established, specific solutions are being introduced to build a safe chain of life for the nanostructures, from their synthesis to their material processing, and from the assembling of systems to their decommissioning.

Overall, a series of R&D activities related to graphene applications in space domains are still to be explored, and this accounts for the impressive growth of research activities being presently dedicated to this strategic sector, a trend clearly evidenced by our analysis of patents and the scientific literature.

On this basis, graphene is expected to help in reaching very soon the technological levels needed to increase the exploration of the solar system and beyond, and to establish and maintain sites on the Moon, Mars, and other planetary bodies. In order to make such ambitious endeavours feasible, not only is the rapid development of graphene-based technologies needed, but so is the additional support of the related space economy, which can be obtained, in particular, by stimulating not only governments and public industries, but also by promoting private companies’ and investors’ involvement in this sector.

## Figures and Tables

**Figure 1 nanomaterials-13-00680-f001:**
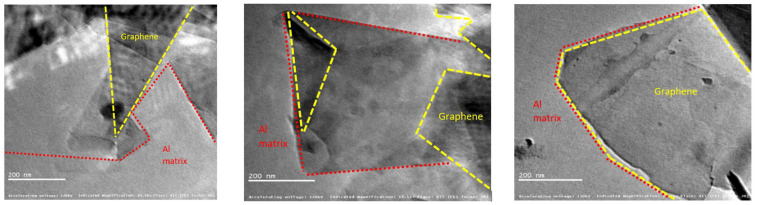
Single-layer graphene inserted in various Al alloy matrixes (reproduced from ref. [22] Copyright© 2022 MDPI).

**Figure 2 nanomaterials-13-00680-f002:**
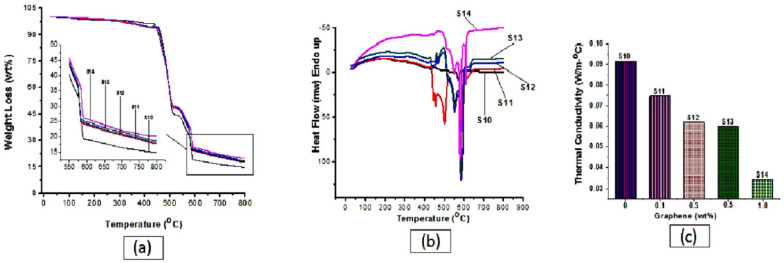
Effect of graphene concentrations on the thermal degradation; (**a**) heat flow response and (**b**) thermal conductivity of (**c**) polymer nanocomposites (reproduced from ref. [27]. Copyright©2018 Trans Tech Publications Ltd., Stafa-Zurich Switzerland).

**Figure 3 nanomaterials-13-00680-f003:**
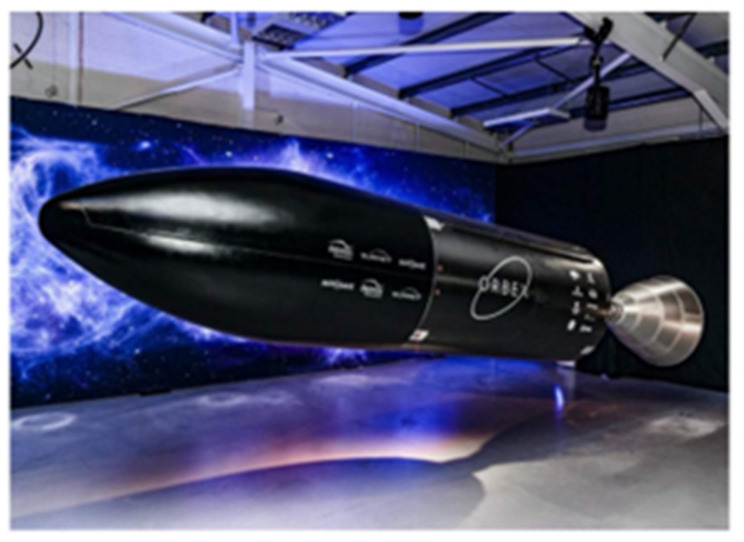
The design of the Orbex Prime body (credits: Orbital Express Launch Ltd., London, UK).

**Figure 4 nanomaterials-13-00680-f004:**
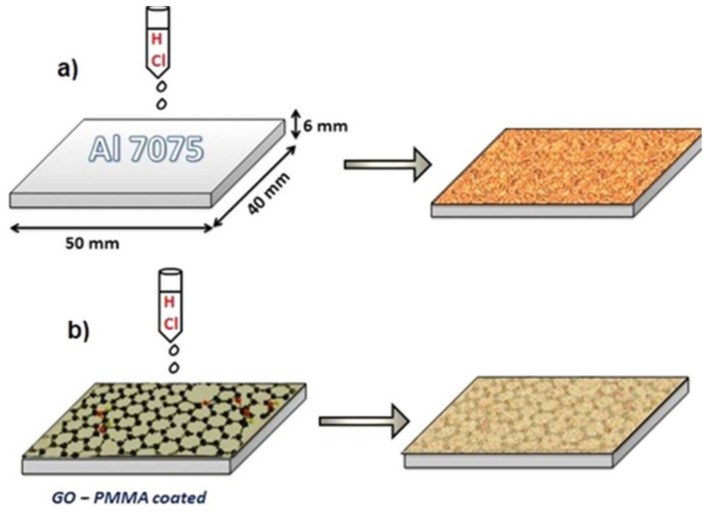
Schematic representation of corrosion behaviour on Al samples; (**a**) corrosion behaviour on the uncoated sample, (**b**) corrosion behaviour on GO–PMMA coated sample (reproduced with permission from ref. [36]. Copyright© 2020 IOP).

**Figure 5 nanomaterials-13-00680-f005:**
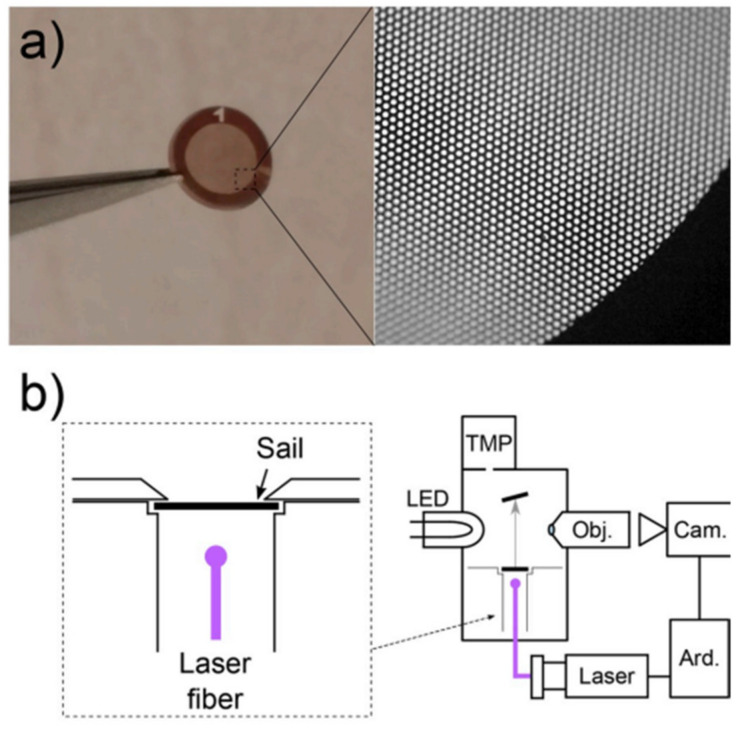
(**a**) Images of a copper grid 3.05 mm in diameter and ~30 μm in thickness. (**b**) Scheme of the system: deposited graphene freestanding over the perforations of the grid, resulting in a sail with light-absorbing circular membranes 6.5 μm in diameter, covering 41% of its area (adapted with permission from ref. [48]. Copyright© 2020 Elsevier).

**Figure 6 nanomaterials-13-00680-f006:**
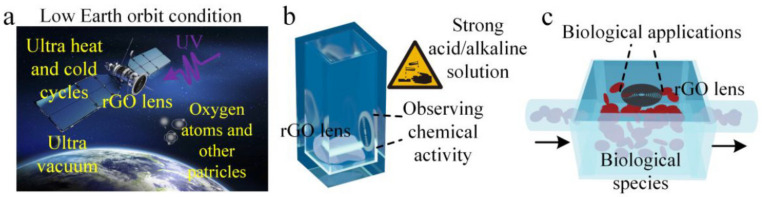
Applications of rGO flat lenses; (**a**) imaging optical element for a satellite in aerospace, (**b**) observing strong acid/alkaline chemical reactions, (**c**) biophotonic microfluidic devices (reproduced with permission from ref [60]. Copyright© 2019 ACS).

**Figure 7 nanomaterials-13-00680-f007:**
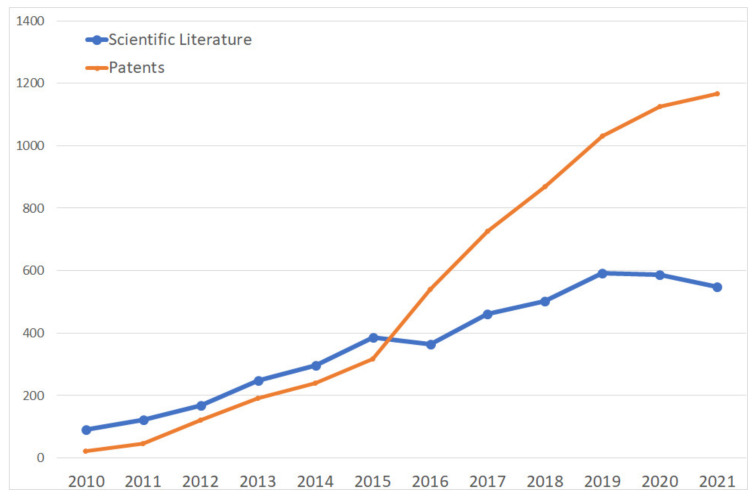
Investment & scientific literature trend for graphene and space applications.

**Figure 8 nanomaterials-13-00680-f008:**
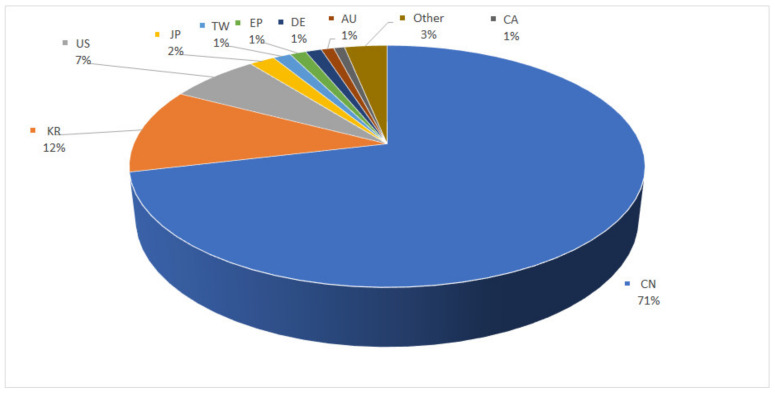
Patents’ first publication country distribution.

**Figure 9 nanomaterials-13-00680-f009:**
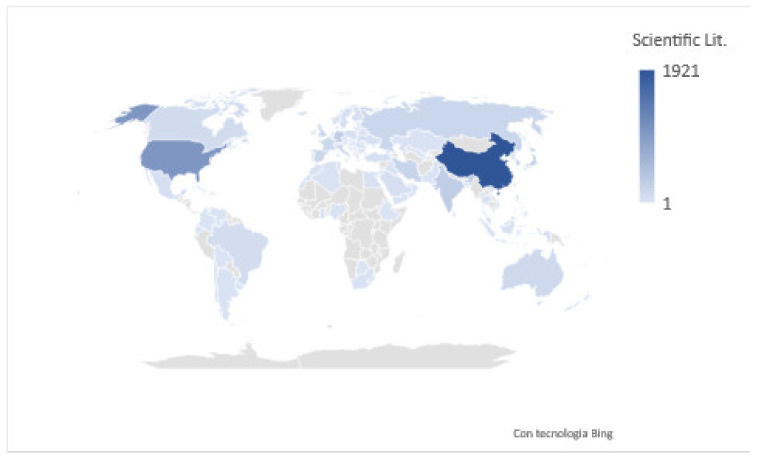
Scientific literature affiliation country distribution (elaborated by the authors using SCOPUS© data).

**Figure 10 nanomaterials-13-00680-f010:**
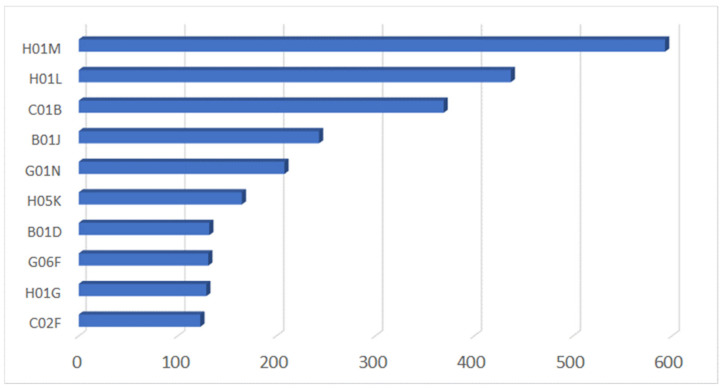
Most frequent IPC4.

**Figure 11 nanomaterials-13-00680-f011:**
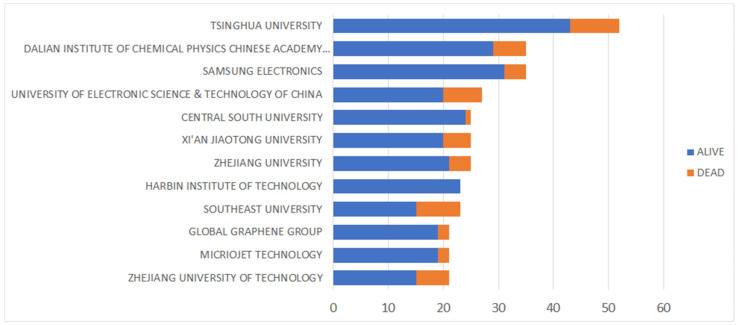
Key assignees by legal state.

**Figure 12 nanomaterials-13-00680-f012:**
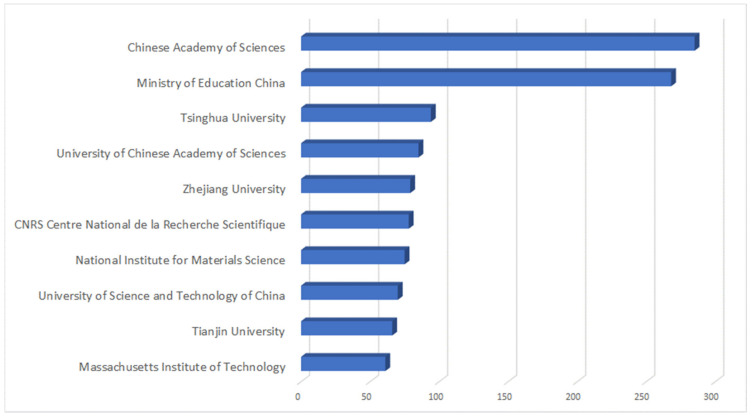
Key affiliations by scientific literature publications.

**Figure 13 nanomaterials-13-00680-f013:**
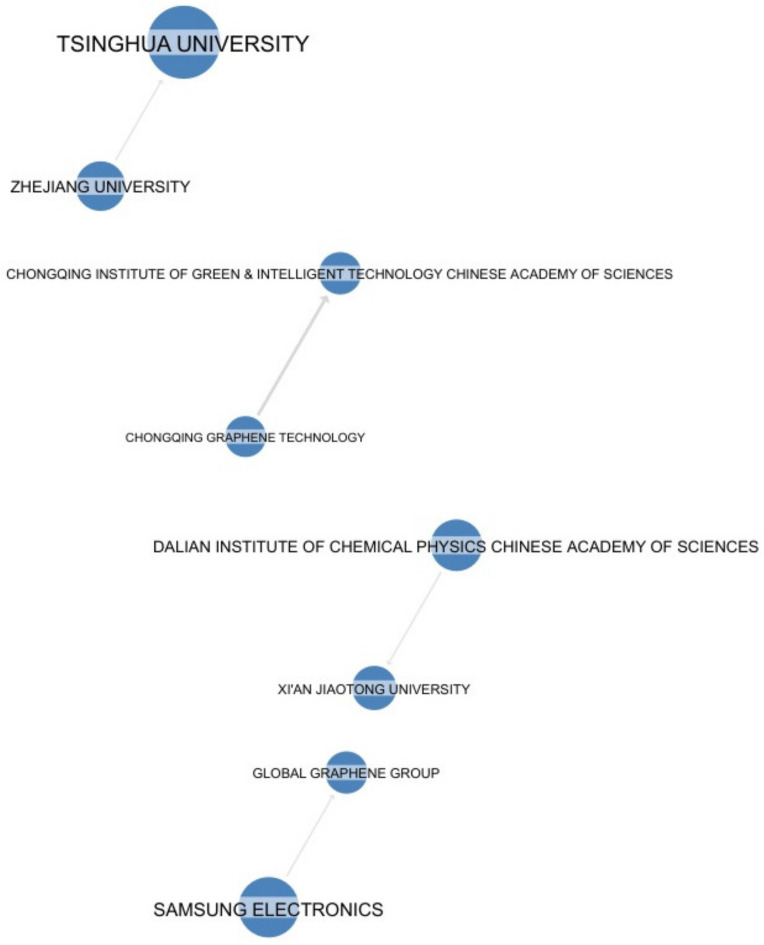
Players’ dependency by citation (©QUESTEL 2022).

**Table 1 nanomaterials-13-00680-t001:** IPC4 descriptions.

IPC4	DESCRIPTION
H01M	Processes or means, e.g., batteries, for the direct conversion of chemical energy into electrical energy.
H01L	Semiconductor devices; electric solid-state devices not otherwise provided for.
C01B	Non-metallic elements; compounds thereof.
B01J	Chemical or physical processes, e.g., catalysis or colloid chemistry; their relevant apparatus.
G01N	Investigating or analysing materials by determining their chemical or physical properties.
H05K	Printed circuits; casings or constructional details of electric apparatus; manufacture of assemblages of electrical components.
B01D	Separation.
G06F	Electric digital data processing.
H01G	Capacitors; capacitors, rectifiers, detectors, switching devices, light-sensitive, or temperature-sensitive devices of the electrolytic type.
C02F	Treatment of water, waste water, sewage, or sludge.

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
