# Peer review of "From Protosolar Space to Space Exploration: The Role of Graphene in Space Technology and Economy"

_nanomaterials, 2023, doi:10.3390/nano13040680_

Round 1

Reviewer 1 Report

This is an interesting review article on the progress of graphene utilization on space-related application. I recommend its publication after revision of the following points. 

-Figure 2 must be reproduced in higher quality, the current version is difficut to read.

-In line 429 there is a typo in the year, it should b 2008 instead of 2000. 

-Use a common style in reference section. 

-It would be interesting to include some comment with the author's view on which domains are the most promising to incorporate graphene, as well point near-future research directions.

-For graphene preparation on metallic and polymer components, as well as for use in supercapacitors, the following articles should be considered for citation:

10.1186/s11671-017-2385-1

10.1016/j.porgcoat.2020.105984

10.1007/s12567-016-0123-7

10.1007/978-94-007-4246-8_2

Reviewer 2 Report

Dear Authors, 

Congrats on your well-written and clear manuscript. Here are some points for you to consider to make it better. 

Shorten the abstract to maybe half the wording. You will lose your audience easily with such long abstracts. Short abstract = higher readership.

Remove the original Fig caption (Fig 18) in your Fig 1.

For the figures, have all copyright been obtained? It should be attributed to as e.g. Reproduced from Ref. XX with permission from publisher XXX.

Do attach the high-definition image of Figure 2. The insert cannot be seen.

One of the possible challenges of using graphene is its toxicity. Maybe you could highlight this in your manuscript?

In line 343, do quotes for all the agencies. 

In figure 7, the caption could be more detailed, maybe adding the keywords used in the search. Do add to the manuscript how was the search done. Is this specific to space applications only?

Fig 8, does not have a legend.

Do attribute the source of Fig 9.

In line 560, do state the keywords of the search. Do cross-reference to the many graphene patent pieces of literature to see if the numbers telly.

For a review paper, 60 references could be too few. It will be great if the authors could include more references (interesting manuscripts related to space/ graphene testing in space).

Good luck!

Round 2

Reviewer 1 Report

accept in present form

Author Response

We have  replaced Fig 6 with a better quality image